# Measurement of the Drug Sensitivity of Single Prostate Cancer Cells

**DOI:** 10.3390/cancers13236083

**Published:** 2021-12-02

**Authors:** Fikri Abali, Narges Baghi, Lisanne Mout, Joska J. Broekmaat, Arjan G. J. Tibbe, Leon W. M. M. Terstappen

**Affiliations:** 1Medical Cell Biophysics Group, MIRA Institute, Faculty of Science and Technology, University of Twente, 7500 AE Enschede, The Netherlands; n.baghi@utwente.nl; 2Department of Medical Oncology, Erasmus MC Cancer Institute and Cancer Genomics Netherlands, Dr. Molewaterplein 40, 3015 GD Rotterdam, The Netherlands; l.mout@erasmusmc.nl; 3VyCAP B.V., Capitool 41, 7521 PL Enschede, The Netherlands; joska.broekmaat@vycap.com (J.J.B.); arjan.tibbe@vycap.com (A.G.J.T.)

**Keywords:** single cancer cells, protein secretion, microwell arrays, drug screening, prostate-specific antigen (PSA)

## Abstract

**Simple Summary:**

Cells communicate mainly through the secretion of proteins. Impaired protein secretion can indicate the development of disease. Cancer cell heterogeneity and acquired resistance to therapy are, however, reducing the effectiveness of cancer treatments. As cancer cells change during the course of the disease, sampling of cancer cells at the time of treatment is needed in order to determine which drugs will be effective. This paper describes a method for measuring secreted prostate specific antigen (PSA) protein from thousands of prostate cancer (PCa) cells. Furthermore, we show that the PSA secretion of individual cells in microwells can be stimulated or inhibited with drugs. To this end, we believe that this method could accelerate the development of new drugs, improve our understanding of resistance to therapy, and, ultimately, improve personalized cancer therapy.

**Abstract:**

The treatment of cancer faces a serious challenge as cancer cells within patients are heterogeneous and frequently resistant to therapeutic drugs. Here, we introduce a technology enabling the assessment of single cancer cells exposed to different drugs. PCa cells were individually sorted in self-seeding microwells, cultured for 24 h, and then exposed to several drugs to induce (R1881) or inhibit (Enzalutamide/Abiraterone) the secretion of a protein (PSA). Cell viability and PSA secretion of each individual prostate cell were monitored over a 3-day period. The PSA protein secreted by each cell was captured on a PVDF membrane through a pore in the bottom of each well. The basal PSA secretion was found to be 6.1 ± 4.5 and 3.7 ± 1.9 pg/cell/day for LNCaP and VCaP, respectively. After exposure to R1881, the PSA secretion increased by ~90% on average and was not altered for ~10% of the cells. PSA production decreased in the majority of cells after exposure to enzalutamide and abiraterone.

## 1. Introduction

Cell-to-cell heterogeneity plays an important role in normal tissue homeostasis and is the driving factor in many diseases [1]. In cancer, the presence of various cell sub-populations represents one of the main challenges to adequate treatment and therapeutic success [1,2,3,4]. If cancer cell heterogeneity can be assessed during the disease course in a non-invasive manner, the optimal treatment for each patient can be chosen at an early stage of the disease. Treatment options for metastatic cancer patients are rapidly increasing [5,6]. As the majority of novel drugs target specific cellular pathways, a thorough analysis of the patient’s tumor is needed in order to select the appropriate drug [4,7].

The heterogeneity of cancer cells and acquired resistance to therapy are, however, reducing the effectiveness of cancer treatments. As cancer cells change during the course of the disease, sampling of cancer cells at the time of treatment is needed in order to determine which drugs will be effective. In patients with metastatic disease, circulating tumor cells can be obtained from blood [8,9]. In cases where the number of these circulating tumor cells (CTCs) is too low, diagnostic leukapheresis (DLA) can be used to obtain a larger number of cancer cells [10,11,12]. However, new tools are needed to assess the cancer cell heterogeneity, determine the effectiveness of therapeutic drugs to be administered, and investigate the mechanisms leading to drug resistance.

Here, we introduce a method that allows for the interrogation of single cancer cells exposed to different drug treatments during cell culture followed by their isolation to determine their genetic composition. To demonstrate the potential of the method, cells from the prostate cancer cell lines VCaP and LNCaP, which produce a prostate-specific antigen (PSA) in response to androgen manipulation, were used. After single cells were distributed in a microwell array, PSA production was measured in response to androgen stimulation and subsequent inhibition by abiraterone and enzalutamide, which are both used in the treatment of castration-resistant prostate cancer (CRPC).

## 2. Materials and Methods

### 2.1. Reagents and Antibodies

Monoclonal mouse anti-PSA (Cat. no. 10-P21A) for the PVDF membrane coating was purchased from Fitzgerald (Fitzgerald Industries International, North Acton, MA, USA). The secondary antibody, a polyclonal rabbit anti-PSA (Cat. No. ab19554), and the detection antibody, an anti-rabbit Alexa 488 (Cat. no. ab150077), were purchased from Abcam (Abcam, Cambridge, UK). Androgen steroid R1881 was purchased from Biotang Inc. (Biotang Inc., Lexington, MA, USA). The inhibitory drugs enzalutamide and abiraterone were purchased from Selleck (Selleckchem, Breda, The Netherlands).

### 2.2. Preparation of the Capture Membranes

Low-fluorescence polyvinylidene (PVDF) membranes with a pore size of 0.45 µm were purchased from Bio-rad (Bio-rad laboratories B.V., Veenendaal, The Netherlands). The PVDF membranes were cut into 1 × 1 cm pieces, wetted in 100% methanol (Fisher Scientific, Leicestershire, UK), and washed three times in sterile phosphate-buffered saline (PBS) (Sigma, St. Louis, MO, USA). Next, the PVDF membranes were coated with monoclonal mouse anti-PSA (25 μg/mL) overnight at 4 °C. Subsequently, the membranes were washed in PBS and blocked in blocking buffer consisting of 3% Bovine Serum Albumin (BSA) (Sigma Aldrich, St. Louis, MO, USA) in PBS for 1 h at RT. The PVDF membranes were washed once in PBS and incubated in a cell culture medium prior to use.

### 2.3. Cell Lines

LNCaP (ATCC CRL-1740) and VCaP (kindly provided by Erasmus MC, Rotterdam, The Netherlands) were grown in RPMI1640 and DMEM (Lonza, Walkersville, MD, USA), respectively, and supplemented with 10% fetal bovine serum (FBS) (Sigma, St. Louis, MO, USA), 100 U/mL penicillin, and 100 μg/mL streptomycin (Lonza, Verviers, Belgium). For passaging, cultures were detached from the culture flask with 0.05% trypsin (Life Technologies Europe B.V., Bleiswijk, The Netherlands) and harvested by thorough pipetting. Subsequently, cells were pelleted by centrifugation and resuspended in fresh complete medium. The cell numbers and viability were determined by incubation of the cells with 4% tryphan blue (Invitrogen, Carlsbad, CA, USA), and the cells were counted on a Luna-FL™ automated cell counter (Westburg B.V., Leusden, The Netherlands). Cells were cultured in T25 treated culture flasks (VWR international B.V, Amsterdam, The Netherlands) in a humidified incubator at 37 °C under 5% CO_2_.

### 2.4. Preparation of the Microwell Array

Microwell chips (VyCAP B.V., Enschede, The Netherlands) were sterilized in 70% ethanol for 60 min and washed in PBS. To remove any ethanol and air in the microwells, the array was placed in a desiccator and a vacuum was applied to allow PBS to enter the wells. To allow cells to adhere and spread onto the silicon nitride of the microwells, the wells were coated with 0.01% poly-L-lysine (Sigma, St. Louis, MO, USA) solution for 1 h at 37 °C. Subsequently, the microwell array was washed with PBS and filled with complete culture medium before cells were seeded.

### 2.5. Capturing PSA from Single Prostate Cancer Cells

To visualize cells after seeding in the wells, they were incubated with 1 mM of calcein AM stain (Invitrogen, Carlsbad, CA, USA) for 30 min or with CellTracker Orange (Invitrogen, Carlsbad, CA, USA) at a dilution of 1:5000 *v*/*v* for 60 min. Labeled cells were spun down and washed twice in culture supplemented with 10% charcoal-stripped FBS (Sigma, St. Louis, MO, USA) to remove any secreted proteins present in the supernatant and avoid cell stimulation. Next, a cell suspension containing around 6000 single cells suspended in medium (supplemented with charcoal-stripped FBS) was distributed into the microwells by applying a small negative pressure of 5 mbar across the microwell chip. Next, the microwells were imaged using an automated inverted fluorescence microscope (Nikon Eclipse Ti2, Tokio, Japan). The device with the cells and the capture surface was incubated at 37 °C and 5% CO_2_ for 24 h (t = 0). After the first 24 h, the membrane was removed and a second anti-PSA-coated membrane was mounted against the bottom of the microwell chip and fresh medium was added that contained R1881 at a concentration of 2 nM. The microwell chip with the membrane mounted on it was again incubated for 24 h at 37 °C and 5% CO_2_ for 24 h (t = 1). The membrane was removed and the microwells were washed with fresh medium before a third antibody-coated membrane was connected to the bottom of the microwell chip and fresh medium with abiraterone (1 μM) or enzalutamide (2 μM) was added and incubated at 37 °C and 5% CO_2_ for 24 h (t = 2). To determine the cell viability after each treatment, cells were stained with calcein AM (live cells) or ethidium homodimer-1 (dead cells).

### 2.6. Detection of Printed Antibody Arrays

For the detection of cell-secreted PSA, the PVDF membranes were incubated once in PBS with 0.05% Tween20 (15 min) and three times in PBS (5 min). Thereafter, membranes were blocked in blocking buffer for 1 h at RT, incubated with secondary rabbit anti-PSA (1:1000 *v*/*v*), and, after three washing steps in PBS (5 min), anti-rabbit IgG (1:1000 *v*/*v*) was added for 1 h at RT to visualize the secreted PSA. Finally, the membranes were briefly washed twice in PBS (5 min) and once in MilliQ water and then dried. The membranes were imaged using an automated inverted fluorescence microscope.

### 2.7. Calibration Curves

One microliter drops of culture medium, containing PSA protein (Cat. no. ab41421) with different concentrations of PSA (0–300 μg/mL), were spotted on the PVDF membrane. After completing the fluorescence labeling of the captured PSA using the procedure described above, the average fluorescence intensity was determined. By measuring the spot size of the spotted medium with PSA, the amount of PSA in pg/µm^2^ was next calculated by dividing the amount of PSA present in the 1 µL drop by the area of the spot. The measured intensity versus the different amounts of PSA in pg/µm^2^ was plotted. Using these calibration values, the fluorescence intensity of the captured spots on the membrane that was mounted against the bottom of the microwell was determined by reading the amount of PSA in pg/µm^2^ and multiplying it by the area of the spot.

### 2.8. Image Acquisition

Images of the microwell array and the fluorescently labeled PVDF membrane were acquired using the imaging function and software of the Puncher system (VyCAP B.V.). This system uses a LED excitation light source, a 20× NA 0.45 objective, and a CMOS camera. We acquired in total 10 × 10 images of the entire 8 × 8 mm surface of the microwell chip and membrane. Images were stored in tiff format without any data compression or loss of signal.

### 2.9. Image Analysis and Quantification of Fluorescent Signals

The location of the spots on the membrane identifies the number of the microwell that contains the cell that secreted the PSA protein. An in-house software algorithm, using Labview’s IMAQ image analysis routines (National Instruments, Austin, TX, USA), determined the location of the same microwell on the different membrane measurements (zero, after stimulation, and after adding therapeutics). The amount of captured protein was next determined by measuring the intensity in each spot and comparing it with the calibration values shown in Appendix A. The Wilcoxon signed-rank test was used for all comparisons (PSA secretion assay). A two-sample *t*-test was used to compare the cell viability after each treatment. A significance level of 0.05 was considered statistically significant; * refers to *p* < 0.05, ** refers to *p* < 0.001 and *** refers to *p* < 0.0001. The data in the dot plots are presented as mean ± SD. The bottom and top of the box are the first and third quartiles, respectively, and the band inside the box is always the second quartile (the median). The square inside the box indicates the mean of all the data.

## 3. Results

### 3.1. Single Cell Seeding and Viability

Figure 1 illustrates the steps used to to detect single cells secreted PSA protein during different treatments inside the microwells. Figure 1a shows a graphical depiction of three of the 6400 wells of the microwell chip. A cell suspension is pipetted onto the top of the chip and a small negative pressure (−5 mbar) is applied across the microwell chip. The fluid enters the wells and exits through the pore in the bottom of the individual wells. The flow-through directs the cells towards the pore and, once a cell lands on the pore, the flow stops and no other cell will enter the same well (Figure 1b). The microwell array functions as a sieve and captures cells based on their size [13,14,15]. To measure secreted PSA, cells from the prostate cancer cell lines LNCaP and VCaP were loaded as single cells into the individual microwells. After the cells were loaded (Figure 1b), the presence of cells in each well was visualized using fluorescence microscopy (Figure 1c). The secreted PSA was captured at the bottom of the microwell array using a poly-vinylidenefluoride (PVDF) membrane that was coated with commercially available anti-PSA antibodies (Figure 1d). The microwell chip with the PVDF membrane was incubated for 24 h to allow secreted PSA molecules to diffuse through the microwell pores towards the membrane (Figure 1e). After incubation, the microwells and membranes were separated (Figure 1f,g). The PSA on the membrane was fluorescently labeled to quantify the PSA at each position and correlated with the position of the microwell (Figure 1h). The microwells were subsequently connected to a second anti-PSA-coated membrane and fresh medium supplemented with the anabolic steroid R1881 was added to stimulate PSA secretion over a period of 24 h (Figure 1i). After 24 h, the membrane was replaced with a third membrane and the cells were supplemented with fresh medium containing an androgen inhibitor (enzalutamide or abiraterone, Figure 1m–p). This procedure resulted in three membranes: one for the detection of PSA from unstimulated cells, one for the detection of PSA after R1881 stimulation, and one to measure the effect of either enzalutamide or abiraterone on PSA secretion. The PSA production and viability of the individual cells exposed to different conditions were monitored in this manner over a 72-h period.

After distributing single cells in the individual wells of the microwell array, images of the cells were acquired to determine their viability and morphological appearance. The design of the well in combination with the microfluidic forces only allow for a single cell per well; however, in cases where two cells are attached to one another, the microwell will contain two cells after the wells are filled [11]. Figure 2a depicts typical images of one or two LNCaP cells inside the wells (blue and red circles, respectively). The cells show a spread morphology similar to that of cells grown in culture flasks. The captured single prostate cancer cells were viable and proliferated inside the microwell for 3 days of culture, indicating that the cell seeding process does not affect cell viability. Figure 2b shows LNCaP and VCaP cells in a microwell at Day 1 and at Day 3 after cell division occurred. After filling the wells, the number of cells per well was quantified. The graph in Figure 2c presents the number of cells per well found for each cell line. The majority of the wells contained a single cell (92%), 4% of the wells contained two cells, 3% of the wells contained three cells, and 2% of the wells contained no cells.

To determine the cell viability with or without drug stimulation, the number of viable cells after each treatment was counted and quantified using the fluorescence of calcein AM and dead cells were detected by the fluorescence of Ethidium homodimer-1. Figure 2d–g show the viability of LNCaP and VCaP cells after different treatments. The viability was first measured for untreated cells left in the microwell chip (t = 24 h). Next, R1881 was added and cells were left for another 24 h before viability was measured (48 h). Finally, one of the drugs (enzalutamide or abiraterone) was added for 24 h and the cell viability was measured once again (t = 72 h). The data show no significant difference in viability between the no-drug group and the androgen-stimulated group (Figure 2d,e). After 24 h, >85% of the cells were viable and >75% of the cells were viable after androgen stimulation (t = 48). However, after the addition of enzalutamide, the cell viability reduced significantly to 24% for LNCaP cells and to 29% for VCaP cells (Figure 2d,f; *p* < 0.0001) when compared with the no-drug group. After the addition of abiraterone, the cell viability was significantly reduced (*p* < 0.0001) to 39% for LNCaP cells and to 46% for VCaP cells when compared with the no-drug group (Figure 2e,g, respectively). In addition, there was also a significant difference between androgen stimulation and androgen inhibition (*p* < 0.001 and *p* < 0.0001). Comparing the same drug on different cell lines also showed a significant difference (*p* < 0.001 for enzalutamide and abiraterone). Exposure to abiraterone resulted in a higher number of viable cells when compared with enzalutamide, with 39% for the LNCaP cells and 46% for the VCaP cells. These results indicate that both cell lines are more sensitive to enzalutamide than to abiraterone, which is in line with previous reports.

### 3.2. Single Cell PSA Secretion Assay

To determine the PSA secretion levels of the cells, a fluorospot assay was developed using a capture membrane coated with antibodies directed against the cell-secreted PSA protein. The secreted PSA molecules captured on the membrane were visualized with fluorescent labeled antibodies directed against PSA. First, we verified that PSA secretion is affected by androgen stimulation (Appendix A). The PSA secretion of prostate cell lines was measured on membranes in 12-well plates and showed the validity of the protocol. Next, we determined whether our microwell platform could be used to capture cell-secreted PSA on PVDF membranes. Cells were distributed into the microwells and their secreted PSA was captured on a PVDF membrane. Figure 3 shows an array of calcein-AM-stained cells trapped in the wells, the corresponding membrane with the captured PSA, and a merged insert showing typical cells and the corresponding secreted PSA. The enlarged view shows high and low amounts of captured PSA corresponding to high-PSA-secreting and low-PSA-secreting cells. To quantify the amounts of PSA captured on the membrane from a well, an image analysis algorithm was developed. First, the images of the cells in the wells and the PSA imprint on the PVDF membrane were overlaid to match the well number with the captured PSA on the membrane. Next, the average fluorescence intensity of the PSA spots was measured. To quantify the PSA production, a calibration curve was plotted to provide the relation between the intensity and the amounts of PSA proteins in pg/cell/day (Appendix A). The background levels were determined by assessment of the fluorescence intensity on the membrane of >50 empty wells across the microwell chip. The results in Appendix A shows that >90% of the single cells secreted PSA at significant higher concentration than the background (empty wells), (*p* < 0.0001). The amount of captured PSA was determined for 1223 wells containing a single LNCaP cell and 329 wells containing two cells. Wells containing one LNCaP cell produced an average of 4.8 pg/cell/day of PSA, which was significantly less compared with wells containing two LNCaP cells with 7.5 pg/cell/day of PSA (Appendix A), (*p* < 0.001).

### 3.3. Drug Sensitivity of PC Cells

To detect the PSA secretion over a 72-h period from cells in wells, membranes were changed every 24 h. Appendix A illustrates the PSA microarray from LNCaP cells at different time intervals without any drug treatment. These results show a dynamic secretion profile and that the procedure for measuring protein over time is reproducible. Once the immunoassays were established, we performed anti-androgen treatments on the cells in the wells to identify their effect on the PSA secretion of PC cells. To do so, we distributed single cells into the microwells as described above. After that, the activated membrane was attached to the bottom of the microwell chip containing the cells. Figure 4 shows the captured PSA spots on three different membranes from the same cells in time. The membrane in Figure 4a shows the PSA secretion from LNCaP cells after 24 h of incubation without the addition of any drug. A large spread in secreted amounts of PSA between individual cells was observed, indicating the existence of a high degree of heterogeneity between the individual cells. Figure 4b shows the PSA secretion profile of the same cells after R1881 stimulation. The cells show an increase in PSA secretion (bigger spots) as well as an increase in the number of secreting cells (17%) with extensive heterogeneity between cells. Figure 4c shows the PSA secretion of the same cells exposed to the anti-androgen drug enzalutamide for 24 h. After the cells were exposed to enzalutamide, the number of viable cells significantly decreased and the PSA secretion of the majority of cells also decreased. To visualize the PSA secretion of individual cells, the same area of the PVDF membrane was enlarged for the three subsequent time points and the secreted products of two cells are indicated with a 1 and a 2 in Figure 4. These results show heterogeneous secretion of PSA by LNCaP cells. The treatment with R1881 induced cells to secrete higher levels of PSA and treatment with the anti-androgen drug enzalutamide showed a decrease in the PSA secretion of the LNCaP cells. Both PC cell lines LNCaP and VCaP show a similar pattern of heterogeneity in PSA secretion.

We then used the fluorescence images to analyze the amounts of PSA protein secreted by PC cells during the three consecutive days. The software algorithm aligned the images and measured the intensities in the captured PSA spots across the three membranes. The calibration curves (Appendix A) enabled the estimation of the amounts of PSA proteins for >1000 stimulated and non-stimulated cells. Using this method, we demonstrated that the cells display extensive heterogeneity (Figure 5a–d). LNCaPs show a distribution of low producers and highly active PSA-producing cells with a large tail (Figure 5a,b, no drug), with an average secretion of ~7 pg/cells/day of PSA protein. Androgen stimulation significantly induced PSA secretion (2-fold) when compared with non-stimulated cells (Figure 5a,b, +R1881, *p* < 0.001). Furthermore, the long tail indicates that more cells are actively secreting PSA. The cells can be grouped into low PSA secretors and a wide distribution of high-PSA-secreting cells. The number of secreting cells decreased when the anti-androgen drug enzalutamide was added (Figure 5a,b). The distribution was more homogenous, the amount of high-PSA-secreting cells decreased, and the average amount of secreted PSA per cell decreased significantly when compared with no drug treatment and R1881 treatment. VCaP cells (Figure 5c,d) show a more uniform secretion pattern when compared with LNCaP cells (Figure 5a,b), with an average of 3.7 pg/cell/day of PSA protein for non-treated cells. The average PSA production level increased significantly (*p* < 0.001) when the cells were stimulated with R1881 to ~4.8 pg/cell/day. Stimulation with anti-androgens significantly reduced the number of cells producing PSA, resulting in a reduction in the median PSA production to <1 pg/cell in VCaP cells (Figure 5c,d, abiraterone or enzalutamide, *p* < 0.001). In addition, there was a difference in the expression of PSA when comparing the effect of the same drug on each cell line (for both drugs, *p* < 0.001).

The percentage of viable LNCaP cells with a PSA secretion level above the background was 53% after 24 h (no drug treatment), 67% after 48 h (R1881 stimulation for 24 h), and 21% after 72 h (abiraterone treatment for 24 h) or 33% after 72 h (enzalutamide treatment for 24 h). The percentage of viable VCaP cells with a PSA secretion level above the background was 30% after 24 h (no drug treatment), 50% after 48 h (R1881 stimulation for 24 h), and 26% after 72 h (abiraterone treatment for 24 h) or 17% after 72 h (enzalutamide treatment for 24 h).

Although we observed a large distribution in secretion activity, clear patterns could be observed. Typical representative cases of the two cell lines are plotted in Figure 5e,h. The majority of the cells responded to stimulation with androgen steroids and increased the PSA secretion levels. In the abiraterone group, LNCaP cells show two populations: androgen-sensitive cells (93–89%, blue lines) and insensitive/low-producing cells (6–11%, purple lines) (Figure 5e,f), while the enzalutamide group shows an added population that is sensitive to androgens but not sensitive to enzalutamide stimulation (<1%) (Figure 5d, red line). The androgen-sensitive cells display a sharp spike in PSA secretion and a decrease following anti-androgen stimulation. The population of androgen-insensitive LNCaP cells, however, did not respond to androgen stimulation and the PSA secretion levels remained low (Figure 5e,f, pink lines), with low variation between assay points.

After the strong increase in PSA production, the PC cells showed a sharp drop in PSA secretion after the addition of abiraterone (Figure 5e,g, blue lines) or enzalutamide (Figure 5f,h, blue lines). Interestingly, a small number of LNCaP cells (<1%) did not show a drop after the addition of the anti-androgen drug enzalutamide and the PSA production remained high (Figure 5f, red line), indicating that the cells required a longer stimulation duration.

## 4. Discussion

To further our understanding of resistance to cancer therapy, tools are needed to interrogate single cancer cells at different times during the course of the disease. This requires methods to isolate and characterize single cancer cells, preferably over time and when exposed to drugs that can potentially be effective. This report presents a platform for the analysis of secreted products from thousands of single cells in individual microwells. Antigens secreted by single PC cells were used to demonstrate the potential of the method. Although single-cell assays can be performed with ELISA or ELISPOT-based methods, our presented single-cell approach provides four advantages over traditional approaches: (a) the ability to handle small samples with few cells; (b) the ability to measure the secreted PSA proteins because they are immunocaptured by the membrane in the vicinity of the cells without being diluted in the culture supernatant; (c) the ability to investigate the cellular heterogeneity of 6400 viable single cells in parallel; and (d) the possibility to retrieve cells of interest for further analysis [13,16]. Because of the hydrodynamic single cell distribution principle, we can efficiently capture 1 cell/well (>90%) and cells remained viable for days. The cells are contained in a well and the products it secretes can be captured on a membrane attached to the bottom of the microwell that can be replaced at any time. Drugs can be added at different time points and the effect on the cell’s viability and the change in the products it secretes measured, followed by its isolation for molecular characterization [11,14].

Although the detection of extracellular PSA in blood is a routine procedure, measuring excreted PSA from single cells is more challenging. Nettikadan et al. [17] developed a protein microarray and was able to detect the PSA amounts secreted from just four LNCaP cells. To increase the sensitivity, Panabières et al. [18] used a combination of a PSA-ELISPOT assay and reported the ability to detect PSA secretion from single LNCaP cells as well as CTCs. In this study, we demonstrated a method to detect secreted PSA from thousands of individual PC cells in parallel. In addition, we show that we can measure the secreted proteins repeatedly at different timepoints as well as stimulate or inhibit the secretion by exposing the cells to different therapeutic drugs. We found extensive heterogeneity in the PSA protein secreted from single cells from the LNCaP and VCaP cell lines. Before exposure to drugs, PSA production was measured in 53% of LNCaP and 30% of VCaP cells with an average of 6.1 ± 4.5 and 3.7 ± 1.9 pg/cell/day for LNCaP and VCaP cells, respectively. Although these are cancer cell lines, the heterogeneity is quite extensive. Moreover, differences in tolerance and the emergence of resistance may be observed upon exposure to drugs. 

We next showed the effect of an androgen steroid or an inhibitory drug on PSA secretion. Treatment with abiraterone significantly decreased the viability of the cells and, for those that survived, the secretion of PSA decreased in both cell lines. Interestingly, a small fraction of the LNCaP cells were not affected by enzalutamide stimulation and their PSA production increased or remained unchanged. Perhaps these cells require a longer stimulation duration at a higher drug concentration in order to elicit a response. As the microwell platform allows for the isolation of these single cells, their molecular profile can be examined and compared with those cells that do respond to the drug to identify mechanisms involved in drug resistance.

In the future, the ability to measure protein activity directly and in combination with other functional responses could constitute a useful new tool for mapping therapeutic efficacy and tumor progression in hormone-sensitive and castration-resistant PC patients. This may help to identify the pathways involved in drug resistance and identify drug candidates that can aid in the management of prostate cancer. In addition, the microwell technology allows for the identification of the molecular mechanism underlying the tolerance to drugs and the potential drug resistance mechanism as the cells can be isolated at any time point for detailed analysis of their nucleic acid composition [13,19].

## Figures and Tables

**Figure 1 cancers-13-06083-f001:**
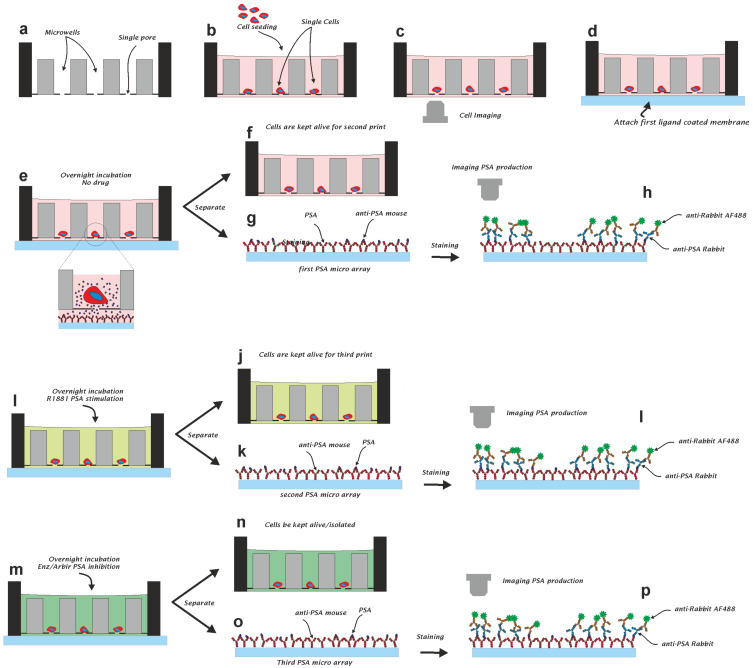
Schematic illustration of the principle used to detect the secretion of protein from single PC cells with or without drug stimulation. (**a**) An array of microwells with a pore in the bottom. (**b**) A single cell suspension is added on top of the microwell array and a negative pressure is applied. (**c**) Images of the cells in microwells are acquired. (**d**) To capture the secreted PSA, a membrane is mounted against the bottom of the microwell array. (**e**) During incubation, cell-secreted PSA is captured on the membrane. (**f**,**g**) Microwells and the membrane are disconnected. (**h**) The membrane is fluorescently labeled to detect PSA and (**i**) microwells are supplemented with a drug and connected to a second membrane. (**j**,**k**) After incubation (24 h), the membrane and microwells are separated. (**l**) The membrane is fluorescently labeled to detect PSA and (**m**) microwells are supplemented with a drug and connected to a third membrane. (**n**,**o**) Finally, after another incubation period (24 h), the membrane is separated from the microwells and (**p**) stained to visualize the membrane that captured PSA from PC cells.

**Figure 2 cancers-13-06083-f002:**
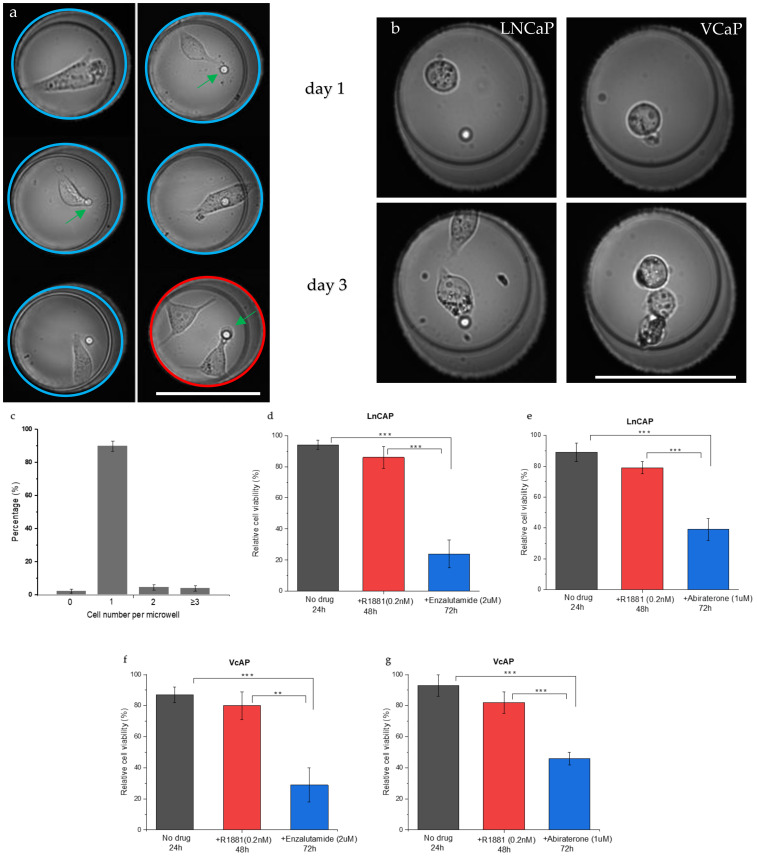
Characterization of the single-cell microwell platform for PSA analysis. (**a**) Typical image of LNCaP cells inside the microwells in which cells adhere to the bottom of the wells and show a spread morphology (**b**). Images showing the proliferation of single LNCaP and VCaP cells after seeding and after 3 days of incubation. The green arrows indicate the 5-µm pores in the bottom of the well. In two of the wells of Figure 2a,b, the cells moved away from the pore and the pore was no longer covered by the cell. (**c**) Characterization of the cell distribution per microwell under optimized cell loading conditions (*n* = 5). (**d**–**g**) Graphs illustrate the viability of the LNCaP and VCaP cells in the microwells with or without drug stimulation. The number of live cells (calcein AM) and dead cells (Ethidium homodimer-1) was counted after each treatment and the cell viability was calculated (*n* = 3). Significance levels are indicated with, ** *p* ≤ 0.001, or *** *p* ≤ 0.0001 (Scale bars: 75 µm).

**Figure 3 cancers-13-06083-f003:**
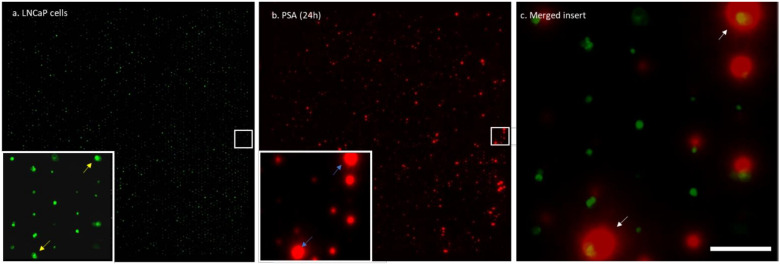
Microwell printing of basal cell secreted PSA. (**a**) Typical example of fluorescently (calcein AM) stained single LNCaP cells sorted in individual microwells. (**b**) The corresponding PSA microarray of the cells captured on a membrane after 24 h of incubation. The white squares show an area of the microarray in which the image is larger and placed in the left lower corner. The yellow arrows in panel (**a**) show typical viable cells and the blue arrows in panel (**b**) show the corresponding PSA signal captured on the membrane. The white arrows in the merged image (**c**) show the cells and the corresponding produced PSA. Scale bar: 100 µm.

**Figure 4 cancers-13-06083-f004:**
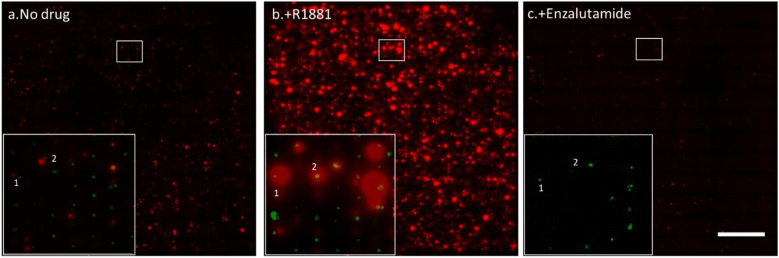
Effect of drug stimulation on PSA secretion. (**a**) Fluorescence image showing the captured PSA of the non-treated cells (no drug). (**b**) Second PSA array from the same cells, stimulated with an androgen steroid (R1881). (**c**) Third membrane with PSA spots secreted from the same cells treated with an anti-androgen drug (enzalutamide). The zoomed-in area of the image at the lower left corner shows a small part of the membrane-bound PSA protein in red and the corresponding cells in green. The numbers 1 and 2 depict a high-PSA-secreting and a low-PSA-secreting LNCaP cell, respectively. Scale bar: 500 µm.

**Figure 5 cancers-13-06083-f005:**
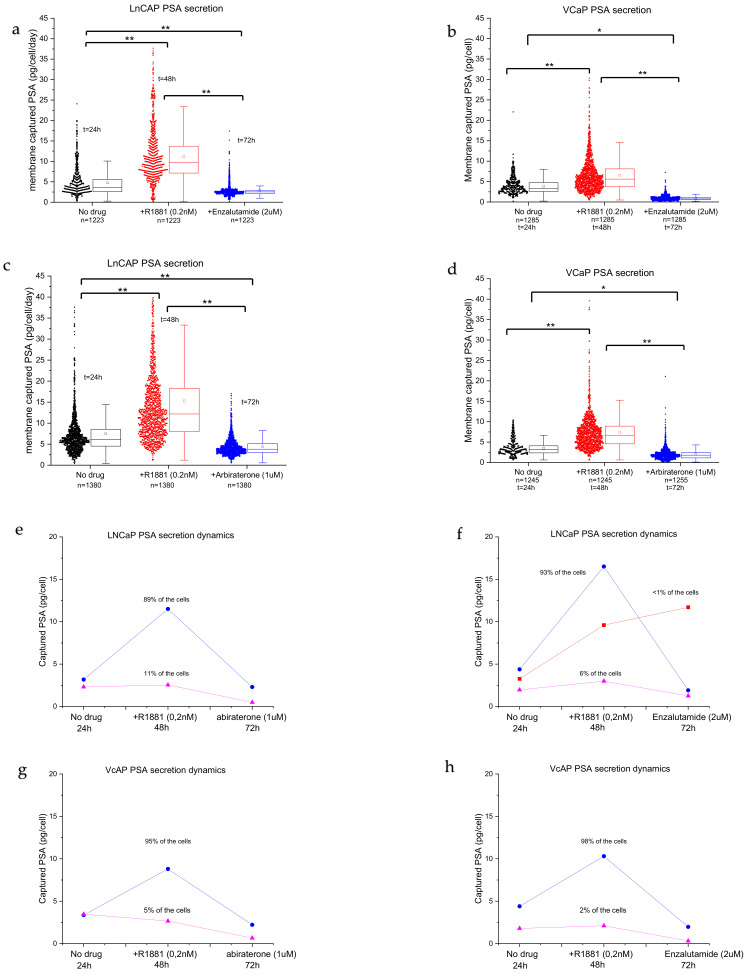
Scatter plot of the PSA secretion from LNCaP and VCaP cells with and without drug stimulation. (**a**,**b**) LNCaP cells with no drug, R1881 stimulation, and (**a**) abiraterone or (**b**) enzalutamide treatment, respectively. (**c**,**d**) VCaP cells with no drug, R1881 stimulation, and (**c**) abiraterone or (**d**) enzalutamide treatment, respectively. (**e**–**h**) Based on the secretion patterns (high and low PSA activity), cells were grouped to reveal different populations. (**e**,**f**) Typical secretion dynamics of representative high activity and low activity PSA-producing LNCaP cells. (**g**,**h**) PSA secretion dynamics of representative high activity and low activity PSA-producing VCaP cells. Significance levels are indicated with * *p* ≤ 0.001 and ** *p* ≤ 0.0001.

## Data Availability

The data presented in this study are available on request from the corresponding author.

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
