# Peer review of "Measurement of the Drug Sensitivity of Single Prostate Cancer Cells"

_cancers, 2021, doi:10.3390/cancers13236083_

Round 1

Reviewer 1 Report

The study is well conducted and manuscript is well written. 

The authors demonstrated a novel method to determine PSA secretion. Only minor  typing errors are in the manuscript. 

Author Response

  • The authors demonstrated a novel method to determine PSA secretion. Only minor typing errors are in the manuscript. 

We have proofread the manuscript and corrected any typos.

Reviewer 2 Report

This manuscript describes a novel technical approach to determine the secretion of proteins at the single cell level. The procedures are adequately described and there are no particular criticisms from the experimental point of view. Some minor points should be addressed:

- Page 2, row 61: some details about antibodies (catalog number, clone code for monoclonal anti-PSA) are missing in materials and methods;

- Page 7, row 224 (Figure 2 caption): “after 24 hours of treatment with R1881 (t=48h) and 24 hours of treatment with enzalutamide or abiraterone (t=72h)”. Shouldn't it be: "and 48 hours of enzalutamide or abiraterone (t=72h)"?

- Page 8, row 231: it cannot be concluded that the two cell lines are more sensitive to enzalutamide than to abiraterone based on that single data, particularly considering that the two compounds were administered at two different doses (2 uM and 1 uM, respectively);

- Figures 3, 4 and S4: it is recommended to add the merge in false colors of the images representing the same fields to better define the correspondence between the spots generated by the single cells at different time points and under different treatments;

In general, the level of English should be improved. There are several poorly worded phrases or repetitions especially in the abstract, simple summary and introduction.

Author Response

This manuscript describes a novel technical approach to determine the secretion of proteins at the single cell level. The procedures are adequately described and there are no particular criticisms from the experimental point of view. Some minor points should be addressed:

  • Page 2, row 61: some details about antibodies (catalog number, clone code for monoclonal anti-PSA) are missing in materials and methods;

The corresponding catalog numbers for each of the antibodies used to bind secreted PSA as well as the antibodies to detect PSA are added to the materials and methods section ‘Reagents and Antibodies’.

  • Page 7, row 224 (Figure 2 caption): “after 24 hours of treatment with R1881 (t=48h) and 24 hours of treatment with enzalutamide or abiraterone (t=72h)”. Shouldn't it be: "and 48 hours of enzalutamide or abiraterone (t=72h)"?

We agree that the sentence can lead to misinterpretation and have changed it to:

“Figure 2d-g shows the viability of LNCaP and VCaP after different treatments. The viability was first measured for untreated cells left in the microwell chip (t=24h), next R1881 is added and cells were left for another 24h before viability was measured (48h) and finally one of the drug (enzalutamide or abiraterone) was added for 24h and viability was measured once again (t=72h).”

  • Page 8, row 231: it cannot be concluded that the two cell lines are more sensitive to enzalutamide than to abiraterone based on that single data, particularly considering that the two compounds were administered at two different doses (2 uM and 1 uM, respectively);

The doses administered were chosen based on the effect of the drug on PSA protein levels and the optimal dosis for our system was chosen based on elipsot titration experiments for detecting secreted PSA at the single cell level.  

We agree that we cannot compare the two groups due to difference in drug concentration. Therefore, this sentence is removed and instead we compared the effect of the same drug on different cell lines. In the revised manuscript we have provided statistical analysis of the data. 

  • Figures 3, 4 and S4: it is recommended to add the merge in false colors of the images representing the same fields to better define the correspondence between the spots generated by the single cells at different time points and under different treatments.

We agree and have revised the figures in the manuscript accordingly.

  1. Figure 3 is updated with a merge image showing the cells their corresponding secreted PSA protein.
  2. Figure 4 is updated with images showing the secreted PSA and the corresponding cells.
  • In general, the level of English should be improved. There are several poorly worded phrases or repetitions especially in the abstract, simple summary and introduction.

We agree and have revised the manuscript accordingly with an emphasis on the abstract, simple summary and introduction  

Reviewer 3 Report

This manuscript by Abali et al., shows a technology enabling the assessment of single cancer cells exposed to different drugs. Based on available literature , I find this study redundant and it doesn't add much to the field. There is no statistical analysis in any of the graphs in the figures.

Author Response

  • This manuscript by Abali et al., shows a technology enabling the assessment of single cancer cells exposed to different drugs. Based on available literature, I find this study redundant and it doesn't add much to the field.

We do not agree that this study is “redundant” as this is the first time a technology is introduced which allows for the simultaneous identification of immunophenotype and corresponding secretome of individual cells permitting the administration of drugs and measuring the response of the individual cells. Currently studies are ongoing in which the technology is applied using CTCs from metastatic prostate cancer patients which we believe can lead to true personalized treatment of this disease.

  • There is no statistical analysis in any of the graphs in the figures.

We have added statistical analysis to the graphs and figures.

Round 2

Reviewer 3 Report

In my opinion, authors must clarify what was the novelty and significance of their research. Also, their findings are insufficient for the Cancers journal. The authors have only shown this technology in vitro, whereas technology platforms already exist for testing multiple drugs in patient derived xenograft models, published few years ago. Therefore, this study is redundant and of low interest.